Land use management based on multi-scenario allocation and trade-offs of ecosystem services in Wafangdian County, Liaoning Province, China

Zhao Wenzhen 1
Han Zenglin 1 2
Yan Xiaolu xlyan@lnnu.edu.cn 2 3
Zhong Jingqiu 2
1 Liaoning Normal University, Urban and Environmental college , Dalian , Liaoning Province , China
2 Liaoning Normal University, Center for Studies of Marine Economy and Sustainable Development , Dalian , Liaoning Province , China
3 Chinese Academy of Sciences, Institute of Applied Ecology , Shenyang , Liaoning Province , China
Costanza Robert
Electronic publication date: 2019 Sep 16
Publication date: 2019
Volume: 7
Electronic Location ID: e7673
Received 2019 May 3; Accepted 2019 Aug 14
Copyright: ©2019 Zhao et al.
Copyright year: 2019
Copyright holder: Zhao et al.
License: This is an open access article distributed under the terms of the Creative Commons Attribution License, which permits unrestricted use, distribution, reproduction and adaptation in any medium and for any purpose provided that it is properly attributed. For attribution, the original author(s), title, publication source (PeerJ) and either DOI or URL of the article must be cited.
License URL: https://creativecommons.org/licenses/by/4.0/

Keywords: Trade-off, Ecosystem services, Land use, Ecological management

Funding: National Natural Science Foundation of China 41571122 Natural Science Foundation Guidance Program of Liaoning Province 20180551194 Education Commission of Liaoning Province of China H201783631 WQ2019020 Social Science Federation 2019 Economic and Social Development of Liaoning Province 2019lslktjd-014 Liaoning Planning Fund Project of Philosophy and Social Science L17CJY003 PhD Research Startup Foundation of Liaoning Normal University BS2018L007 This work was supported by the National Natural Science Foundation of China (41571122), the Natural Science Foundation Guidance Program of Liaoning Province (20180551194), the Education Commission of Liaoning Province of China (H201783631, WQ2019020), the Social Science Federation 2019 Economic and Social Development of Liaoning Province (2019lslktjd-014), the Liaoning Planning Fund Project of Philosophy and Social Science (L17CJY003), and the PhD Research Startup Foundation of Liaoning Normal University (BS2018L007). The funders had no role in study design, data collection and analysis, decision to publish, or preparation of the manuscript.

==============================
Developing effective methods to coordinate the trade-offs among ecosystem services (ES) is important for achieving inclusive growth and sustainable development, and has been the focus of scholars and ecosystem managers globally. Using remote sensing and geographic information system (GIS) data, our study examined Wafangdian County of Liaoning Province as a case study to reveal the spatiotemporal evolution of four ES (food supply [FS], net primary productivity [NPP], water yield [WY], and soil conservation [SC]) and changes among their interactions. Then, an ordered weighted averaging model was introduced to simulate the optimal scenario of ES allocation. Results showed that: (1) the spatial and temporal changes in ES were significant over 14 years. All ES presented an inverted U-shaped growth curve from 2000–2014. (2) Synergies were observed within provisioning services, and there were trade-offs between provisioning services and regulating services, as well as provisioning services and supporting services. (3) The optimal scenario for Wafangdian was scenario 5 (trade-off coefficient, 0.68). The allocation of FS, NPP, WY, and SC in scenario 5 were 0.187, 0.427, 0.131, and 0.063, respectively. Implementing each ES weight of optimal scenario in land use management contributed to achieving intercoordination of ES. We propose to coordinate land and sea management to restore natural habitats that were expanded into in the high ES area. It is our anticipation that this study could provide a scientific basis for optimizing the allocation of ES and improving land use structure of coastal zones in the future.

Introduction

Ecosystem services (ES) can be considered to be natural capital assets, which provide various benefits to human society (Daily & Matson, 2008; Zheng et al., 2019). Managing ES is intended to achieve both environmental sustainability and socioeconomic benefits (Bohnke-Henrichs et al., 2013; Granek et al., 2010). Its essence is to enhance regional integration of ES through science-based management, which is also the purpose of China’s “Ecological Civilization Construction” (Li, Zhu & Xiao, 2019). The Millennium Ecosystem Assessment of the United Nations classifies these services into four categories: provisioning services, regulating services, supporting services, and cultural services (Ahmed, 2002). The specific interactions among different ES are complex (Barbier et al., 2008; Brauman et al., 2007); however, they can be abstracted as competing trade-offs and synergies of mutual gains (Dai et al., 2016; Rodriguez et al., 2006). They have become hot topics in geography and ecology to understand these trade-offs and synergies among various ES; to identify their types, characteristics, and spatiotemporal patterns; and to maximize the benefits to human society from multiple interconnected ES (Fu & Yu, 2016; Queiroz et al., 2015; Rodriguez et al., 2006).

Land use change is one of the most important factors affecting ES by changing the structure, processes, and functions of the ecosystem (Ceschia et al., 2010; Chen et al., 2019). Human intervention in land use is an effective way to manage negatively affected in ES. Under increasing human development, the depletion of natural resources and rapid changes in land use have resulted in unbalanced ES (Zheng et al., 2019). In the past decade, ES trade-offs have been widely evaluated and studied, and have been reported to occur between provisioning services (e.g., agricultural production) and regulating services (e.g., soil conservation, carbon storage), as well as between cultural services (e.g., aesthetic, recreation) and provisioning services (Lorilla et al., 2018; Rodriguez et al., 2006; Zheng et al., 2019). There is no doubt that the trade-offs among ES will continue to intensify owing to the increasing demand for natural resources from an increasing human population (Alcamo et al., 2005). In response to such global trends, scientific land use management can minimize the loss of alternative services and facilitate the management of ES (Schroter et al., 2014; Wong et al., 2015). However, previous research on land use management focused on case descriptions and analysis based on trade-offs and synergies of ES (Butler et al., 2013; Lorilla et al., 2018; Rodriguez et al., 2006). Few studies have focused on the allocation and trade-off of ES. Effective methods for transforming scientific understanding of ES trade-offs into policy are still lacking. Ordered weighted averaging (OWA) can effectively reflect the effect of decision-makers’ subjective perception on evaluation results by simulating different preference decision-making scenarios combining order weight and criterion weight (Yager, 1988). Previous studies have demonstrated the ability of the OWA model to balance the internal conflicts among ES, and it has been widely used in land suitability appraisals, geological disaster prevention, and ecological security assessments (Malczewski & Rinner, 2005; Delgado et al., 2013; Zhang et al., 2015). However, few studies have applied OWA to land use decision management based on the ES allocation and trade-off.

Wafangdian is at the node of the Northeast Asia Economic Circle. After 2010, the Changxing Island Economic and Technological Development Zone and Taiping Bay Port Area were established successively, and a large-scale reclamation campaign was launched. In addition, industrial effluent from petrochemical factories has polluted coastal waters. In particular, the establishment of Hengli Petrochemical Company, the world’s largest PTA petrochemical company, has exacerbated this threat. Therefore, the present study was carried out in Wafangdian County with the aim to explore land use management based on ecosystem services. Specifically, the objectives of our study were: (1) to evaluate the four typical ES and their interrelationships, (2) to optimize ES allocation and trade-off based on the OWA-GIS model, and (3) to recommend land use management strategies. Our research can not only further our understanding of the complex interactions among typical ES but also provide scientific reference for land use planning for decision makers and stakeholders.

Figure 1 Location of Wafangdian County in China.

Materials & Methods

Study area

The ES analysis was developed for Wafangdian County of Liaoning Province (121°13E–122°16′E, 39°20′N–40°07′N), in northeast China (Fig. 1). Wafangdian covers 3,881 km2 and its western region is near the Bohai Sea with a coastline of 461.2 km. Altitude ranges from sea level to 800 m. With its beneficial geographical position and strong industrial foundation, Wafangdian has become the leading county-level economy in Liaoning Province, and is known as the “bearing capital” of China. However, the geographical advantages of land and sea not only bring great opportunities for development but also cause ecological problems for Wafangdian. As for the marine ecological environment, with the advancement of macrocosm urbanization strategies, urban construction and industrial development in Wafangdian turn to the coastal zone, and the ecological pressure increases year by year. Natural ecosystems, such as estuaries and beaches, have been encroached on a large scale. The sewage discharged from petrochemical factories into the sea has deteriorated water quality in sea area, and the disorderly development of mariculture has gradually reduced the ecological balance of the sea. In addition, terrestrial ecological problems are also serious. A large number of chemical enterprises in Dalian have been transferred to aggravate air pollution. Furthermore, seasonal drying of major rivers increases the pressure on the water supply. Wafangdian urgently needs to co-ordinate land and sea, strengthen land use management, and improve integrated ES.

Data source

The study data are from 2000, 2007, and 2014. Five data types were used in this study: (1) the land use data of the three years were interpreted from remote sensing images (Landsat7 ETM and Landsat8 OLI) with a spatial resolution of 30 m. According to the actual situation of the study area, the land use types of Wafangdian were interpreted into seven categories: forest, grassland, cropland, construction land, inland water, mariculture area, and unused land. (2) The normalized difference vegetation index (NDVI) data were obtained from the MODIS13Q1 dataset on the NASA website (https://www.nasa.gov/), with a spatial resolution of 250 m. Bilinear interpolation was used to improve the resolution accuracy of the NDVI to 30 m to ensure that all raster data was at the same resolution. (3) A 30-m resolution digital elevation model was provided by Geospatial Data Cloud Site, Computer Network Information Center, Chinese Academy of Sciences (http://www.gscloud.cn/). (4) The meteorological data were collected from the China Meteorological Data Network (http://data.cma.cn/). We obtained monthly precipitation, evaporation, and average temperature; and total and net solar radiation pertaining to Wafangdian and 13 surrounding counties. ArcGIS 10.2 software was used to interpolate the monthly meteorological data by inverse distance weighting, so that the meteorological point data could be converted into 30 m resolution raster data. (5) Soil property data came from the China Soil Map-Based Harmonized World Soil Database version 1.1 (http://westdc.westgis.ac.cn).

Quantifying ecosystem services

Ecological problems should be considered, such as water shortage and soil erosion caused by rapid urbanization in Wafangdian. At the same time, the government’s decision-making tendency in dealing with ecological problems is also considered. Therefore, this study selected four ES that were representative of the study area, including two provisioning services (food supply [FS], water yield [WY]), one regulating service (soil conservation [SC]), and one supporting service (net primary productivity [NPP]). The mapping and quantification of ES was implemented for 2000, 2007, and 2014, and results are presented on a grid map with 30-m resolution.

Food supply: food supply is an important provision services of ecosystems, especially agricultural ecosystems. Based on the land use classification, this study uses the yield per unit area of different land use types as an indicator to measure the FS capacity of each land use type. There is a significant linear relationship between the yield of crops and livestock products and NDVI (Li, Guo & Peng, 2012; Zhao et al., 2012). Therefore, NDVI (range 0–1) was used to spatialize the statistical data of food yield. Specifically, the grain yield and husbandry yield were allocated to cropland and grassland grids, respectively, according to the NDVI. The calculation formula used was as follows: (1) Pi=NDVIiNDVIsum×Psum

where Pi refers to the grain yield and animal husbandry at grid i. Psum represents the gross grain yield and animal husbandry. NDVIi is the NDVI at grid i; NDVIsum is the sum of NDVI of cropland or grassland in the study area. In addition, the yield of freshwater aquaculture and marine aquaculture was assigned to inland waters and marine aquaculture areas, respectively.

Net primary productivity: NPP is an important index for evaluating vegetation productivity and coverage and is an important component of the terrestrial carbon cycle (Field et al., 1998). It was used to express the organic matter energy accumulated per unit area of vegetation in a unit time; i.e., the residual amount of organic matter created by deducting oxygen respiration from vegetation during photosynthesis (Ruimy, Saugier & Dedieu, 1994). In our study, NPP was modelled using the process-based Carnegie–Ames–Stanford approach (CASA) (Potter et al., 1993). Two important variables were used to account for NPP in the CASA model: the absorbed photosynthetically active radiation (APAR) and the actual solar energy utilization efficiency of plants ( ε). (2) NPPx,t=APAR,t×εx,t

where APAR (x,t) the photosynthetically active radiation (MJ/m2 ⋅ a) absorbed by the pixel x in the month t. ε(x, t) refers to the actual utilization rate of light energy at the x pixel in the month t.

Water yield: WY was calculated using the water production module of the InVEST model v.3.3.3. Considering the influence of topography on runoff and the spatial differences in soil permeability under different land use types, the InVEST model is based on the principle of water balance to estimate the water supply of different landscape types quantitatively (Hoyer & Chang, 2014). The basic principles are as follows: (3) Yx=1−AETxPx×Px

where Y(x) is the annual WY of grid x (mm); AET (x) is the actual annual evaporation of grid x (mm); P (x) is the annual precipitation of grid x (mm).

Soil conservation: Soil conservation was simulated using the revised universal soil loss equation (RUSLE) model (Renard et al., 1997). Based on the replacement hypothesis of surface cover, SC amount can be expressed as the difference between potential soil erosion and actual soil erosion; i.e., the difference between the amount of soil erosion without any vegetation cover management and soil and water conservation measures and the amount of soil erosion under current vegetation cover management and soil and water conservation measures (Wu et al., 2017). The calculation formula used was as follows: (4) Ac=Ap−Ar

where Ac represents SC (t/km2 ⋅ a), Ap is the amount of the potential soil erosion (t/km2 ⋅ a), and Ar is the amount of the actual soil erosion (t/km2 ⋅ a). The potential soil erosion and actual soil erosion were quantified as follows: (5) Ap=R×K×LS

(6) Ar=R×K×LS×C×P

where R is the rainfall erosion factor; K is the soil erosion index; LS represents the combined slope length (L) and factor of slope length (S); C is the vegetation cover index; and P is the SC measure index.

Correlation analysis among ecosystem services

To quantify the relationships among the four ES (FS, NPP, WY, and SC) we calculated the correlation coefficient matrix among ES (Raudsepp-Hearne, Peterson & Bennett, 2010; Turner et al., 2014). Each indicator was standardized by the maximum value ranging from 0–1 to remove the effect of ES unit differences. We used ArcGIS 10.2 to extract standardized ES into tables using Create Fishnet tools, and then calculated the Spearman’s rank correlation for all pairs of ES using the scatterplot matrix function of the “car” package in R 3.4.3 (Fox & Weisberg, 2011; R Core Team, 2017). Moreover, correlation coefficients were tested in SPSS 20.0 (SPSS Inc., Chicago, IL, USA).

Based on the correlation analysis, the trade-offs among ES were clarified. When a correlation coefficient among ES passed the significance test at 0.01 level and the coefficient was negative, then there was deemed to be a significant trade-offs relationship. On the contrary, if the correlation coefficient was positive, then there was a significant synergies relationship (Jopke et al., 2015). In our study, all the significant correlation coefficients were classified into strong correlations (|r| ≥ 0.5), moderate correlations (0.3 ≤ |r| < 0.5), and weak correlations (0.1 ≤ |r| < 0.3) according to the classification of the intensity of significant correlation coefficients of Cohen (1992).

Scenario analysis based on OWA-GIS

Ordered weighted averaging is a method for controlling factor weight combination. By reordering the data of each index according to their attribute value and giving different order weights and weighting aggregation values according to the order of each index, OWA reflects the decision-making results when the decision maker ranks the importance of each index differently (Yager, 1996; Zhang, Wang & Song, 2017). Its model principle is as follows: (7) OWAxij= ∑i=1nwisij,wi∈0,1and∑inwi=1,foriandj=1,2,3…n

where xij refers to four standardized raster maps of ES (NPP, SC, FS, and WY) sij is the new sequence obtained by reordering the attribute values of xij in descending order. In our study, we ranked the new sequence according to the average value of each ES; wi was the ordered weights of the sij; n refers to the number of ES.

We introduced the OWA operator into the trade-offs analysis of ES and simulated decision-making scenarios under different preferences by setting different risk coefficients. At the same time, combined with the calculation of trade-offs degree (Zhang et al., 2015), the best trade-offs of ES in Wafangdian under feasible decision-making conditions was determined. The specific steps were as follows:

(1) Four evaluation indicators (four ES) were ranked in descending order according to their mean value.

(2) The order weights w and trade-offs degree with different risk-degree coefficients were calculated (Yager, 1996; Zhang et al., 2015) as follows (Table 1): (8) wi=QRIMin−QRIMi−1n,i=1,2,3…n

(9) QRIMr=rα,α∈0,+∞

(10) trade−off=1−n ∑inwi−1n2n−1,0≤tradeoff≤1

Where wi is the order weight; QRIM is a monotone increasing function; i is the ordinal number, and n represents the number of ES; r is the independent variable; and α is the risk coefficient (scenario), also known as the optimistic degree coefficient, which based on the risk perception of decision-making caused by the difference in the numerical value of indicators and the difference in subjective weights.

Table 1 Weight and trade-offs under different scenarios.

ω1: soil conservation (SC); ω2: food supply (FS); ω3: water yield (WY); ω4: net primary productivity (NPP).

Scenario	Risk coefficients (α)	ω1	ω2	ω3	ω4	Trade-off	
1	0.0001(α → 0)	1.000	0.000	0.000	0.000	0.000	
2	0.1	0.871	0.062	0.039	0.028	0.170	
3	0.5	0.500	0.207	0.159	0.134	0.630	
4	1	0.250	0.250	0.250	0.250	1.000	
5	2	0.063	0.187	0.313	0.437	0.680	
6	10	0.000	0.001	0.055	0.944	0.070	
7	10,000 (α → ∞)	0.000	0.000	0.000	1.000	0.000	

(3) Combining the OWA operator with the GIS platform, the OWA-based ES raster was obtained.

(4) Using the Getis-Ord Gi* module in ArcGIS 10.2 to analysis the cold and hot spots of the OWA-based ES raster under different scenarios. In our study, 99% of the hot spots were defined as the highest ES area, 95% as a high ES area, and 0 as a medium ES area. On the contrary, 99% of the cold spots were named as the lowest ES area, and 95% were low ES area. Finally, the trade-off coefficient and the area of the highest and high ES areas of each scenario were combined to determine the optimal trade-offs of ES in Wafangdian.

Figure 2 Mapping of ecosystem services in 2000, 2007, and 2014.

NPP: net primary productivity; FS: food supply; WY: water yield; SC: soil conservation; (A) FS in 2000; (B) FS in 2007; (C) FS in 2014; (D) NPP in 2000; (E) NPP in 2007; (F) NPP in 2014; (G) WY in 2000; (H) WY in 2007; (I) WY in 2014; (J) SC in 2000; (K) SC in 2007; (L) SC in 2014.

Results

Overall change in ecosystem services

The maps in Fig. 2 show the distribution of ES in 2000, 2007, and 2014. FS was highly distributed in the central and western areas but had a smaller distribution in the east. The high value areas were mainly in the vast central cropland and southwestern coastal waters, and the overall trend decreased from west to east. The average FS yield in 2000 was 173.33 t/km2. In comparison, the mean FS yield in 2007 was 250.57 t/km2, with a 44.56% increase compared with 2000. The mean FS production was 228.17 t/km2 in 2014 and decreased by 8.94% compared with 2007. In terms of temporal scale, FS yield showed a trend of rapid growth in the early stage followed by a slow decline in the later stage. Among them, the FS capacity increased significantly in the southwest coastal waters and the cropland along the lower reaches of the Fuzhou River.

The total NPP of Wafangdian in 2000, 2007, and 2014 was 1.62 trillion g C, 1.85 trillion g C, and 1.65 trillion g C, respectively (Table 2), showing a similar trend to that of FS in temporal scale. However, the spatial distribution was different from that of FS. The high value areas were mostly concentrated in the mountains with high vegetation coverage, such as Laomao Mountain and Longtan Mountain in the east and Dabei Mountain in the south-central. Low value areas were mostly distributed in the southwest coastal areas, and in Fuzhou Town and the downtown of Wafangdian.

Table 2 Average values of ecosystem services (ES) in 2000, 2007, and 2014.

NPP: net primary productivity; FS: food supply; WY: water yield; SC: soil conservation.

ES	2000	2007	2014	Change rate (%)	
				2000–2007	2007–2014	
FS (t/km2⋅ a)	173.33	250.57	228.17	44.56	−8.94	
NPP (g C/m2⋅ a)	412.42	478.83	423.05	16.10	−11.65	
WY (mm)	230.20	390.02	197.35	69.43	−49.40	
SC (t/km2⋅ a)	61.46	66.28	48.03	7.84	27.53	

The spatial pattern of WY was mainly influenced by precipitation and evapotranspiration. It was higher in the west compared with the east, in the aquatic habitats compared with the terrestrial ones, and in the flat lands compared with the mountains. The spatial distribution of WY was similar in 2000, 2007, and 2014; however, the average annual WY varied considerably among the 3 years. In the early stage, it increased from 230.20 mm in 2000 to 390.02 mm in 2007, and then decreased sharply to 197.35 mm in 2014.

The average SC in 2000, 2007, and 2014 was 61.46 t/km2, 66.28 t/km2, and 48.03 t/km2, respectively (Table 2), showing a moderate increase and then a sharp decrease through time. The spatial distribution of SC was similar to that of NPP. Forest dominated the mountainous area in the northeast, with high vegetation coverage; therefore, the SC had a high capacity. The terrain in the southwest was flat, where towns and villages were concentrated. Owing to the high intensity of land development and utilization, the SC capacity was low. On the whole, except for the sustained growth of SC, the ES of the study area increased initially and then decreased from 2000 to 2014.

Spatiotemporal changes in ecosystem services with different land use types

Our study focused on the changes in ES and their relationships among forest, grassland, and cropland (three vegetation-covered land use types). The mean values of the four ES according to each land use type were determined. To ensure a more intuitive comparison among the different ES, min–max normalization was used to remove dimension and then Nightingale’s rose diagram was made. As shown in Fig. 3, compared with cropland and grassland, forest had the largest NPP and SC, while grassland had the largest FS. The WY of cropland was the highest, while SC and NPP were the lowest, and FS was slightly lower than that of grassland. In terms of temporal dynamics, the WY and NPP of the forest increased and then decreased from 2000 to 2014 (NPP: 0.68 in 2000, 0.77 in 2007, and 0.62 in 2014; WY: 0.23 in 2000, 0.29 in 2007, and 0.12 in 2014), while SC increased. However, the change was not significant (SC increased by 0.01 in 3 years). The WY of the grassland also presented an inverted U-shaped curve from 2000 to 2014 (0.32 in 2000, 0.35 in 2007, and 0.23 in 2014), while FS and NPP decreased significantly (FS: 0.79 in 2000, 0.71 in 2007, and 0.67 in 2014; NPP: 0.56 in 2000, 0.43 in 2007, and 0.50 in 2014). Except for SC, the other three ES in cropland showed a significant inverted U-shaped trend representing an initial increase that then decreased over the 14 years investigated (Fig. 3). Overall, our results showed that NPP, FS, and WY decreased in forest, farmland, and grassland from 2000 to 2014. SC in the three land use types increased slightly; however, this was not significant (Fig. 3).

Figure 3 Rose map of ecosystem services in 2000, 2007, and 2014.

NPP: net primary productivity; FS: food supply; WY: water yield; SC: soil conservation; (A) ES rose map of forestland in 2000; (B) ES rose map of grassland in 2000; (C) ES rose map of cropland in 2000; (D) ES rose map of forestland in 2007; (E) ES rose map of grassland in 2007; (F) ES rose map of cropland in 2007; (G) ES rose map of forestland in 2014; (H) ES rose map of grassland in 2014; (I) ES rose map of cropland in 2014.

Figure 4 Spearman’s correlation coefficient matrix between ES in 2000 (A), 2007 (B), and 2014 (C).

Red numbers indicate positive correlation and blue numbers indicate negative correlation; ** correlation is significant at the 0.01 level.

Trade-offs and synergies

Figure 4 shows the Spearman’s correlation coefficients and significance levels among ES in 2000, 2007, and 2014. These values were used to describe trade-offs and synergistic relationships. During the 3 years, all 18 pairs of ES passed the significance test at 0.01, with seven pairs strongly correlated (—r— ≥ 0.5), three moderately correlated (0.3 ≤ |r| < 0.5), and eight weakly correlated (0.1 ≤ |r| < 0.3). From 2000 to 2014, except for NPP and SC, FS and WY were synergistic, and the other pairs of services were trade-offs. In the synergistic relationship, NPP and SC had the highest synergistic relationship, with a significant positive correlation (2000: r = 0.52; 2007: r = 0.62; 2014: r = 0.53), while FS and WY had a moderately positive correlation in 2000 (r = 0.30) and weakly positive correlation in 2007 and 2014 (2007: r = 0.10; 2014: r = 0.12). Among the 12 pairs of trade-offs, that between NPP and WY was the highest, with a significantly negative correlation (2000: r =  − 0.63; 2007: r =  − 0.67; 2014: r =  − 0.64). The trade-off between NPP and FS in 2014 was the weakest, with a weak negative correlation (r =  − 0.08). Overall, during the study period, trade-offs dominated the relationships among ES in Wafangdian.

To clarify the trends in trade-offs and synergies among ES in 2000, 2007, and 2014, we drew a histogram of the correlation coefficient of each pair of ES (Fig. 5). The two pairs of synergistic ES changed significantly from 2000 to 2014. The synergy between NPP and SC increased and then decreased in 14 years, while the synergy between WY and FS decreased sharply over the same period, with a drop of 46.15%. For the four pairs of trade-offs, NPP-WY and SC-WY both presented inverted U-shaped curves over 14 years, while SC-FS decreased from 2000 to 2007, and then increased from 2007 to 2014. However, the trade-off of NPP-FS decreased consistently over 14 years without fluctuation. The results showed that the trade-offs and synergies between ES without human intervention were more highly significant and their changes were more stable (synergy: NPP-SC; trade-offs: NPP-WY and SC-WY). Conversely, ES directly affected by human intervention, such as FS, tended to change considerably and irregularly (trade-offs: NPP-FS and SC-FS; synergy: FS-WY). In addition, the trade-off between regulating services and provisioning services was intense and stable, especially the relationship between regulating services and WY, which remained stable without fluctuating (SC-WY: NPP-WY).

Figure 5 Comparison of correlation coefficient among ecosystem services in 2000, 2007, and 2014.

NPP: net primary productivity; FS: food supply; WY: water yield; SC: soil conservation.

Optimal allocation scenario of ecosystem services and land use

Through setting seven different levels of risk coefficients, and according to the three formulas (Eqs. (8)–(10)), the trade-offs and weights under each scenario were determined. Figure 6 shows the OWA-based raster layer of ES under the seven scenarios based on the weights in Table 1. The OWA-based ES spatial distribution was different under each scenario. Compared with the other six scenarios, the ES of scenario 1 presented low values. The spatial distribution of the high and low values of scenarios 2–4 were more uniform than those of scenario 1. However, the high values were more concentrated in the coastal mariculture areas, while the ES of terrestrial areas were distributed evenly but were still dominated by low values. The trend in high value clustering of ES under scenarios 5–7 became increasingly evident. In particular, there were large sea–land differences under scenarios 6 and 7. The high values were mostly concentrated in the eastern mountainous areas covered by forest, while the coastal waters were completely covered by low values. Overall, from scenario 1–7, the spatial distribution of OWA-based ES tended to be low-value clustering to uniform distribution to high-value clustering.

Figure 6 The spatial distribution of ecosystem services based on ordered weighted averaging-graphical information system (OWA-GIS) model under different scenarios.

(A): Scenario 1 (α = 0.0001); (B): scenario 2 (α = 0.1); (C): scenario 3 (α = 0.5); (D): scenario 4 (α = 1); (E): scenario 5 (α = 2); (F): scenario 6 (α = 10); (G): scenario 7 (α = 10,000).

Scenarios 1 and 7 were two extreme scenarios (extremely optimistic and pessimistic, respectively) in decision making, which were dominated by a single type of ES. These two scenarios do not exist in reality and also do not meet the needs of diversification of ES. Therefore, we excluded scenarios 1 and 7, and then the cold hot spot analysis module in ArcGIS 10.2 was used to determine the high (hot spot) and low (cold spot) value aggregation under the other five scenarios (Fig. 7). For the spatial distribution of ES regionalization, scenario 2–6 changed from low value agglomeration dominant (scenario 2) to coexisting high-value and low value agglomeration (scenario 6). Under scenario 5, the high ES areas radiated to the western plains and to the coastal areas, including the highest ES observed. The smallest sea–land differences in ES were observed under all scenarios.

Figure 7 Ecosystem services (ES) regionalization under different risk scenarios.

(A):Scenario 2 (α = 0.1); (B): scenario 3 (α = 0.5); (C): scenario 4 (α = 1); (D): scenario 5 (α = 2); (E): scenario 6 (α = 10).

The results of ES regionalization (Table 3) showed that the sum of the high ES areas under scenario 5 covered an area of 2,086.50 km2, accounting for 53.91% of the total area of Wafangdian, far exceeding that of the other scenarios. Combining the trade-off coefficients of each scenario in Table 1, scenario 5 had a trade-off coefficient of 0.68, which was the highest under all feasible decision-making scenarios. Although the trade-off coefficient of scenario 4 was 1, the weights of services were equal under this scenario, which is an ideal decision-making scenario for the complete equalization of ES. By comparing with the spatial distribution and area of the high ES areas of each scenario and combining with the trade-off coefficients of each scenario, it was found that scenario 5 had the largest high ES area (53.91% of the whole region) and the highest ES trade-off coefficient (0.68). Consequently, the optimal trade-off scenario in Wafangdian was the area defined under scenario 5.

Table 3 Characteristics of ecosystem services (ES) regionalization for scenarios 2–6.

(A): Highest ES area; (B): High ES area; (C): Medium ES area; (D): Low ES area; (E): Lowest ES

Scenario	Area of each ES Regionalization (km2)	Proportion of A and B	
	A	B	C	D	E		
2 (α = 0.1)	22.57	121.73	656.68	2,229.73	839.26	0.04	
3 (α = 0.5)	226.80	1,190.46	950.71	945.73	556.27	0.37	
4 (α = 1)	257.06	1,195.97	943.30	940.28	533.36	0.38	
5 (α = 2)	686.20	1,400.30	1,113.45	465.50	204.51	0.54	
6 (α = 10)	362.20	702.44	1,765.16	410.27	630.37	0.28	

Figure 8 shows that, in the optimal trade-off scenario, the high and highest ES areas of Wafangdian covered all kinds of land use types and were distributed throughout the study area. Cropland encompassed an area of 730.28 km2 and constituted 35% of all of the high ES areas. Forest and mariculture area were ranked second and third, respectively. Forest encompassed an area of 542.49 km2 and constitutes 26% of all of the high ES areas.

Figure 8 Land use structure under the optimal scenario.

(A) Spatial distribution of high and highest ES area; (B) land use structure under the optimal scenario.

Discussion

Ecosystem services mapping and drivers of variation

In the present study, for the purpose of making better decisions on the optimization of environmental and social services, methods based on remote sensing data were used for spatially visualizing ES to enable stakeholders and policy-makers to compare with the actual land use (Mina et al., 2017). Therefore, three ES use an evaluation model based on remote sensing or GIS (SC based on RUSLE, WY based on InVEST, and NPP based on CASA). Maps were comparable after standardization, which also facilitated the subsequent implementation of map algebraic overlay and GIS spatial analysis. The maps produced by the models were the products of a linear combination of multiple variables, such as evapotranspiration, soil composition, vegetation cover index, and even statistical data of grain yield, which serve to supplement and enrich land use data. The information provided by combinatory ability of the model was more accurate and reliable than that obtained from the direct interpretation of land use (e.g., Eigenbrod et al., 2010; Kandziora, Burkard & Muller, 2013).

The map in Fig. 2 shows the spatial distribution of the four ES analyzed. The two provisioning services (FS and WY) maintained the same spatial distribution pattern. This was different from the results of Sun, Ren & Zhao (2016), whose results suggested that, where FS is high, the water consumption of crops is also high, and thus, the higher the FS the lower the WY. These results are reasonable for inland areas or research areas excluding offshore waters. However, the large area used for aquaculture, and the highly important position of fisheries in the local economy, introduce difficulties when attempting to classify coastal aquaculture areas into land use types. The high production value and yield of mariculture areas result in a broader distribution of FS in aquatic areas compared with that of terrestrial areas.

From 2000 to 2014, all ES presented a fluctuating inverted U-shaped curve, and 2007 was the node for all ES fluctuations when each ES reached its highest level in 14 years. Meteorological conditions, such as precipitation, temperature, and radiation, were the drivers of the observed fluctuations. The monthly average precipitation, average temperature, and total solar radiation in 2007 were higher than those in 2000 and 2014 of the same period. Precipitation in 2007 was particularly high with monthly average precipitation increasing by 74.31% compared with that of 2000 and twice as much as was observed in 2014 (Figs. S1 and S2 and Table S1). These meteorological factors played a decisive role in the CASA, water yield module of InVEST, and RUSLE models (Huston, 2012), which directly contributed to increases in ES, such as NPP, WY, and SC, in 2007. In addition, our results clearly demonstrate that the particular hydrothermal conditions in 2007 resulted in a higher crop yield in that year compared with those of the same period over 14 years. Especially for Wafangdian, which is located in the temperate monsoon climate zone, timely and sufficient precipitation and heat in spring and summer (April–September) ensure the growth of crops and other plants.

Interaction among ecosystem services

The specific interactions between different ES were complex. The results of correlation analysis among ES in different research areas are not consistent, especially regarding the relationships among multiple ES in some areas where the interface between humans and land is more prominent, which have not been clarified (Wu et al., 2017). In addition, the trade-offs has spatiotemporal heterogeneity, and the synergies determined in one region is often expressed as a trade-off in another region. For example, the study by Enfors et al. (2008) at catchment scale showed that FS and SC are synergistic, while a study by Maes et al. (2012) across the European continent showed a trade-off between these two ES. Therefore, the trade-offs and synergies among ES derived from our study are not globally uniform, or even nationally, and are also different from the results of previous studies.

Our study identified the synergies within provisioning services, the trade-offs between provisioning services and regulating services, and the synergies between supporting services and regulating services in Wafangdian. Previous studies have reported the synergies between WY and regulating services, but trade-off between WY and FS (Qiu & Turner, 2013; Sun, Ren & Zhao, 2016). Conversely, our results revealed a more differentiated picture. One of the reasons for the difference is as mentioned in the analysis of WY spatial distribution pattern in ‘Ecosystem services mapping and drivers of variation’; i.e., differences in study site location. Another reason is the different models of WY estimation. Sun, Ren & Zhao (2016) used a method considering comprehensive water-holding capacity to calculate WY. This method estimates WY by calculating canopy interception, water retention in litter layers, and loose soil interception (Wang, 2007). The mechanism of this estimation is more focused on reflecting the capacity of water regulation, which is essentially a regulating service. Similar to most previous studies, our results indicate trade-offs among provisioning services and other services (Schirpke et al., 2019; Jopke et al., 2015). This was especially evident for the two pairs of trade-offs involving WY (WY-NPP and WY-SC), which were the most significant among all trade-offs in this study. Synergistic effects were often found in the same type of ES (Raudsepp-Hearne, Peterson & Bennett, 2010; Zhao et al., 2018), which is further demonstrated by the moderate synergy observed between FS and WY in our study. In addition, similar to the results of Schirpke et al. (2019), our results showed that the synergies between supporting services (NPP) and regulating services (SC) are all determined by the presence of forest owing to its high capacity for ES provision.

As shown in Fig. 5, the changes in ES trade-offs in Wafangdian in the 14 years investigated were complex, and there were either increasing or decreasing trade-offs and synergies among ES. Therefore, it is difficult to judge whether the change in overall ES allocation was positive. In addition, although trade-offs and synergies have changed to some extent during the study period, there was no mutual transformation between trade-offs and synergies observed in our study. Thus, it can be concluded that there were no reversible trade-offs in Wafangdian in the 14 years investigated. Evidently, a pair of trade-offs (NPP-FS) and a pair of synergies (WY-FS) involving FS were both weakened in the 3 years, which contrasts with the observed decreasing trend in cropland. Whether the advancement of farming technology promotes the growth of grain production or whether the continuous expansion of offshore mariculture compensates for the decline in land FS capacity remains to be further investigated.

Ecosystem service optimization and land use management strategies

The basic concept of the OWA-based trade-off coefficient of ES is different from the trade-off among ES. Optimal trade-offs refer to the allocation of ES that maximize the service capacity of an ecosystem (Zhang et al., 2015). The optimal trade-off coefficient based on OWA considers the risks of decision-making and the value of each service to the combined provision of services to redistribute the existing service allocation. In summary, the optimal trade-off coefficient is essentially an optimization objective for the status quo of regional trade-offs and synergies.

In the optimal scenario (scenario 5), NPP and WY were found to be highly capable ES that exhibited stable changes over the 14 years, and thus were highly weighted, while highly fluctuating FS and SC with low ES capability were designated as lower weight. Therefore, to optimize the interactions among ES and improve the overall ES level, it is entirely reasonable to increase the weight of services with stable ecological functions and decrease that of services with large fluctuations (Liao et al., 2018). In the future, land use management strategies should consider the eastern mountainous areas and western coastal areas as important areas to focus on and implement ecosystem protection and restoration projects. On the one hand, from the perspective of strengthening existing protected areas, the eastern area should strengthen the management of forest resources, and achieve the goal of returning farmland to forests at the edge of forested lands, steep-sloped areas, water-source areas, and highly-polluted cultivated lands in the eastern area contain Lao Mao Mountain, Longtan Mountain, and Camel Mountain. On the other hand, the western coastal area is more seriously affected by ES degradation. Relevant departments should strengthen the protection of typical ecosystems such as coastal wetlands, estuaries, and bays, and also important fishery waters such as spawning grounds, bait grounds, wintering grounds, and migration channels. They should establish wetland nature reserves in the estuary area of Fuzhou Bay, and restore wetland ecosystems in the Fuzhou and Fudu estuaries. Such management strategies that coordinate land–sea habitats can help to maintain a balanced and efficient operation of ES in Wafangdian.

Human intervention in ecosystems should fully consider the resistance and recovery of different ecosystems to external disturbance (Bertness, Brisson & Crotty, 2015). Therefore, the resilience of different ecosystems should be coupled in formulating Ecosystem Optimization objectives. The four ecosystem services provided by Wafangdian forestland have changed steadily in 14 years, while coastal wetland and cropland have changed dramatically. Most previous studies showed that forest ecosystem had great natural recovery and ecology resilience, while wetland and farmland ecosystem were more fragile and less resilient (Liu, Gao & Wang, 2018; Seidl, Rammer & Spies, 2014). Evidently, there was a linear correlation between ecosystem services and ecosystem resilience (Li, Zhou & Wu, 2017). The study of ecosystem resilience is of great significance to the ecological risk warning and sustainable management of ecosystems (Bertness, Brisson & Crotty, 2015). Detecting and analyzing the fluctuation of ecosystems regime shift, correcting and constraining the objectives of Ecosystem Optimization with different ecosystem resilience values are more conducive to achieving sustainable management of regional ecosystems, which also require continuous monitoring and regulation in follow-up research.

Conclusions

Coordinated land–sea management of ES in coastal cities has always been a challenge for scientific and effective land use management. Our study revealed that the GIS-based OWA multi-criterion evaluation method can be used to determine the optimal allocation of ES trade-offs by adjusting land use artificially. The results showed that ES in Wafangdian fluctuated over the 14 years tested, which is reflected in the increase from 2000–2007 and the decrease from 2007–2014. Under the influence of ES changes, the trade-offs and synergies also fluctuated. Based on the trade-offs and synergies in 2014, an optimized service allocation is proposed; i.e., the ES allocation under scenario 5. In the future, ES research should systematically determine the trade-off and synergy mechanisms and promote the application of the results in the land field. At present, China is actively seeking solutions to balance economic development and ecological protection, such as land use planning and ecological red line delimitation. Research into trade-offs and synergies can effectively identify the ecological space that is essential for maintaining regional ES and requires protection.

Supplemental Information

Figure S1 Monthly precipitation and temperature of Wafangdian in 2000, 2007, and 2014

Click here for additional data file.

Figure S2 Monthly total solar radiation of Wafangdian in 2000, 2007, and 2014

Click here for additional data file.

Table S1 Variation of climatic factors in different years of Wafangdian

Click here for additional data file.

Supplemental Information 1 Meteorological data for evaluating NPP and water yield including precipitation (mm), evaporation (mm), temperature (°C), and solar radiation (J/m2)

Click here for additional data file.

Supplemental Information 2 Key parameters for evaluating water yield based on InVest model

Click here for additional data file.

Supplemental Information 3 Local soil data for assessing water yield and soil conservation including soil type, soil depth, soil composition, organic matter content, etc

Click here for additional data file.

Supplemental Information 4 The water yield data of the sub-basin simulated by InVest model

Click here for additional data file.

Supplemental Information 5 The standardized data of ecosystem services using the maximum and minimum method, which is used to calculate the trade-off between ecosystem services

Click here for additional data file.

We would like to thank Editage for English language editing.

Additional Information and Declarations

Competing Interests

Author Contributions

Data Availability

The authors declare there are no competing interests.

Wenzhen Zhao conceived and designed the experiments, performed the experiments, analyzed the data, contributed reagents/materials/analysis tools, prepared figures and/or tables, authored or reviewed drafts of the paper.

Zenglin Han and Xiaolu Yan conceived and designed the experiments, performed the experiments, analyzed the data, approved the final draft.

Jingqiu Zhong contributed reagents/materials/analysis tools, prepared figures and/or tables.

The following information was supplied regarding data availability:

The raw data are available in the Supplemental Files.

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
