# Peer review of "Land use management based on multi-scenario allocation and trade-offs of ecosystem services in Wafangdian County, Liaoning Province, China"

_PeerJ, doi:10.7717/peerj.7673_

## Round 0.1 · original submission · Major Revisions

You need to do a better job of explaining the methods and results in language accessible to a broad audience and address the other reviewer comments.

·

Basic reporting

I am not familiar enough with the many methods used in this paper to provide a substantial review. It is my sense that this study used data from only two years (2010 and 2014) to determine changes to ecosystem services that are sensitive to rainfall. NPP was determined to grow in this time span and is described has having a 1.7% increase despite lower rainfall in 2014. The Food Supply (FS) ecosystem service increased over this time period primarily (as far as I understood) through increases in aquacultural activity. Food supply was reported in Yuan per hm. I saw no rationale for reporting food supply in monetary units which are subject to many market forces that have nothing to do with ecosystem services. I am also not familiar with hm as an areal unit (Hectameter ?).

The discussion of trade-offs and synergies between ecosystem services was particularly confusing. It seemed to be explained solely by correlations between changes in ES using only 2010 and 2014 as temporal observations with sample size needs coming from doing this on a per pixel basis? In addition - changing prices of grain were used to explain changes in ecosystem services. I do not understand that. A simple example (ginned up for simplicity): A yield of 1 ton per hectare brings $1,000 yuan in 2010 while a 0.5 ton per hectare brings in $10,000 yuan in 2014 is an INCREASE in FS? How does that make sense? I do not believe there is sufficient temporal data in this analysis to justify any conclusions whatsoever with respect to change or allocation strategies.

Experimental design

Not enough temporal data?

Validity of the findings

As I said in the beginning - there are many methods used here that I am not familiar with. However, I am concerned about the limited temporal resolution of this experimental design. I am also perplexed by the idea of grain price changes influencing changes to the food supply ecosystem service. I would be more comfortable with calories of food yielded. This would be confusing also if major changes were made to agriculture e.g. switching from grains to livestock or orchards. etc.

Additional comments

This paper needs a broad overview of Wafangdian County. Not necessarily a description of latitude and longitude but a description of the ecosystems that exist there, variability of weather and climate in this area, impacts of water availability and climate on these ecosystems, the economic activity going on in this area, and changes that have taken place to all of these from 2010 to 2014. I suggest placing your data and analysis in that context. The ESs evaluated were FS (derived from NDVI alone?), NPP (derived from CASA in which you saw a 1.7% increase - what may have caused that? Land use management? What evidence is there for that? less rainfall?), WY (Using InVEST - what data changed from 2010 to 2014 that went into that other than differences in annual precipitation? Why do you believe the results of this model were valid when applied to your data?), and SC (using RUSLE - did anything other than rainfall change in the evaluation of SC for 2010 and 2014?). I am not familiar with the OWA analysis.

Reviewer 2 ·

Basic reporting

no comment

Experimental design

no comment

Validity of the findings

no comment

Additional comments

Study the effects of land use management on the trade-offs of ecosystem services is meaningful to investigate the key factors and process of ES trade-offs change. This study applied GIS, ecological models and n ordered weighted averaging model to examine spatial and temporal changes of four ES (food supply, net primary productivity, water yield, and soil conservation), and identify the effects of land use management on the ES trade-offs based on multi-scenario allocation. It’s meaningful, and I suggest revise several key issues before the acceptance as the following:

1. L66-68, the gaps on studying the effects of land use management on the trade-offs of ecosystem services need to be justified. This section mainly listed the studies on the impact of land use change and management on ecosystem services and trade-offs, but the main problems in the current studies are not clearly concluded, which need to be addressed clearly, to demonstrate the significance of your study.

2. L89-97, the ecological and environmental problems in the study area (Wafangdian) are not explained clearly, which needs to be added to enhance the representativeness of the study area.

3. I don’t understand why there are food supply for the forest and grassland? Please justify it and give a clear definition in the sub-section of Quantifying ecosystem services.

4. In this study, the CASA, InVEST and RUSLE models were used to estimate NPP, WY and SC services. There was no any modifications and calibrations of the models, and the default parameters were straightly used. So it is not necessary to introduce these models at detail. I suggest to explain each model with only one equation and cited related references is enough. Now, it’s too redundant for the explanation of these models.

5. L217-260 of Scenario analysis based on OWA-GIS is also suggested to simplify, and Fig.2 is not necessary.

6. The sub-titles of “Spatiotemporal distribution of ecosystem services” and “Spatiotemporal differences in ecosystem services with different land use types” in the RESUKTS are suggested to revised as “Overall change of ecosystem services” and “Spatial change of ecosystem services”, respectively, and reorganize the related contents.

7. There are too many figures and tables. Some figures/tables are redundant and can be removed, such as Figure 2 and Table 1, etc. Also, the captions of the figures are usually removed from the figures, please have a check for the requirement of the figures for PeerJ. Some figures can be put into the Appendix.

8. The conclusion needs to be further enhanced to a higher level, for example, by addressing the future direction or significance for ecosystem services study.

9. The English and context need to be improved, especially for the sections of Results and Discussions

10. The format of references need to be double-checked and consistent.

Reviewer 3 ·

Basic reporting

The four ES are not all using abbreviations in Lines 37,38,45. I recommend to use abbreviations like Line 139.

Experimental design

The scenario analysis based on OWA-GIS should be described with more details.

Validity of the findings

The conclusion should be better written.

Additional comments

Very interesting to see investigated complex interactions among ecosystem services.This study selected four ES that were representative of the study area and applied OMA model to optimize allocation .The result may provide scientific methods for land use management.
A few questions are list below.
1 How to determine risk coefficients?
2 Please provide more details on fuzzy quantification model (Table 2).

---

## Round 0.2 · Minor Revisions

Just a few additional minor revisions before we accept the paper.

·

Basic reporting

The paper has been significantly revised as a result of the first review. These revisions are acceptable and it is particularly useful that a third temporal dataset (2007) has been added to demonstrate the confounding impact of weather / climate variability on this analysis.

Experimental design

Adding the 2007 dataset was quite useful. I did notice that ‘preserving ecoystem’ diversity was an ‘objective’ in the scenario development. This seems to be a ‘tradeoff’ With respect to optimizing ecosystem services. I think a greater discussion of how ‘optimization’ in terms of efficiency is almost always a tradeoff with another often referred to and desired objective of ‘resilience’. This appeared to be focused on optimizing ES but what happens to the resilience of the system as a result of this optimization. I think a full of analysis of this is beyond the scope of this paper but a few sentences describing this ‘tradeoff’ is warranted.

Validity of the findings

This is a relatively sophisticated analysis of a very complex problem. It is very difficult to assess ‘validity’ in this respect. Assessing the validity of Explorations of tradeoffs and interactions amongst ES using models and datasets that have myriad shortcomings and inaccuracies is virtually impossible. I think the best we can do is publish the findings, and see how they hold up over time. Those that seem to correspond with reality in the future deserve greater attention then. Just as many General Circulation Models of global climate get presented, assessed, and reevaluated.

Additional comments

Thank you for you attention and response to the comments.

Reviewer 2 ·

Basic reporting

Clear English is used!

Experimental design

Research question is well defined.

Validity of the findings

Conclusions are well stated.

Additional comments

The authors have answeres all my concerns. It's suitable to be accepted for publication. Thanks for your responses!

---

## Round 0.3 · accepted · Accept

Thank you for the final revisions.